# The Effect of a School-Based Intervention on Children’s Cycling Knowledge, Mode of Commuting and Perceived Barriers: A Randomized Controlled Trial

**DOI:** 10.3390/ijerph19159626

**Published:** 2022-08-05

**Authors:** María Jesús Aranda-Balboa, Francisco Javier Huertas-Delgado, Patricia Gálvez-Fernández, Romina Saucedo-Araujo, Daniel Molina-Soberanes, Pablo Campos-Garzón, Manuel Herrador-Colmenero, Amador Jesús Lara-Sánchez, Javier Molina-García, Ana Queralt, Diane Crone, Palma Chillón

**Affiliations:** 1PROFITH “PROmoting FITness and Health through Physical Activity” Research Group, Sport and Health University Research Institute (iMUDS), Department of Physical Education and Sports, Faculty of Sport Sciences, University of Granada, 18011 Granada, Spain; 2Teacher Training Centre La Inmaculada, University of Granada, 18013 Granada, Spain; 3IDAF Research Group, Department of Didactics of Musical, Plastic and Corporal Expression, University of Jaen, 23071 Jaen, Spain; 4AFIPS Research Group, Department of Teaching of Musical, Visual and Corporal Expression, University of Valencia, 46021 Valencia, Spain; 5AFIPS Research Group, Department of Nursing, University of Valencia, 46010 Valencia, Spain; 6Centre for Health, Activity and Wellbeing Research, Cardiff Metropolitan University, Cardiff CF5 2YB, UK

**Keywords:** active commuting, cycling, perceptions, adolescents, school

## Abstract

The low rates of active commuting to/from school in Spain, especially by bike, and the wide range of cycling interventions in the literature show that this is a necessary research subject. The aims of this study were: (1) to assess the feasibility of a school-based cycling intervention program for adolescents, (2) to analyse the effectiveness of a school-based cycling intervention program on the rates of cycling and other forms of active commuting to/from school (ACS), and perceived barriers to active commuting in adolescents. A total of 122 adolescents from Granada, Jaén and Valencia (Spain) participated in the study. The cycling intervention group participated in a school-based intervention program to promote cycling to school during Physical Education (PE) sessions in order to analyse the changes in the dependent variables at baseline and follow up of the intervention. Wilcoxon, Signs and McNemar tests were undertaken. The association of the intervention program with commuting behaviour, and perceived barriers to commuting, were analysed by binary logistic regression. There were improvements in knowledge at follow-up and the cycling skill scores were medium-low. The rates of cycling to school and active commuting to/from school did not change, and only the “built environment (walk)” barrier increased in the cycling group at follow-up. School-based interventions may be feasibly effective tools to increase ACS behaviour, but it is necessary to implement a longer period and continue testing further school-based cycling interventions.

## 1. Introduction

Active commuting to and from school (ACS) can be a routine behaviour that enables pupils to be more physically active during their day by walking or cycling. In addition, the World Health Organization (WHO) suggests cycling is a way to achieve daily physical activity (PA) recommendations in the youth population (i.e., 60 min daily of moderate-to-vigorous PA) [1]. Today, low rates of PA and high levels of sedentary behaviour among youth are increasing worldwide [2]. This situation is worrying, given that PA behaviour learned during childhood is transferred to adulthood [3,4]. Moreover, active commuting can provide other benefits, such as improvements in general health [5], including mental health [6], and improving academic and cognitive performance [7]. Furthermore, cycling as a mode of commuting may help to increase levels of cardiorespiratory fitness in young people, reduce obesity, and decrease the risk of cardiovascular disease, diabetes and depression, among other benefits [8,9,10]. In addition, the use of cycling as a mode of commuting may contribute to reduce the use of private cars or other motorized vehicles, avoiding associated health problems from poor air quality [11,12]. Moreover, PA-related health benefits can persist in adulthood, so the promotion of PA from early ages [13,14] is an important social function. The ideal environment identified to promote health-related behaviour, such as active commuting, has been found to be in the school context [15].

However, in recent decades the prevalence of active commuting to school has decreased in several countries, including the United States of America [16], Australia [17], China [18] and Spain [19,20]. Nowadays, cycling to school in some European countries, such as Spain and Ireland is still very low, at around the 2% [21,22]. In order to promote ACS, a variety of studies and programmes have been implemented to increase the prevalence of active commuting behaviour in the youth population [23,24]. In fact, several reviews on interventions promoting walking or cycling to school are available in the scientific literature [25,26,27], concluding that interventions are focused mainly on walking to school and in children. Consequently, Larouche et al. [26] suggested that the implementation of cycling interventions in secondary school settings are necessary to increase cycling and active commuting to schools. However, the literature on the effectiveness of several school-based interventions to promote cycling remains unclear [28,29,30]. For example, Mandic et al. [31] observed that cycle skills training improved children’s cycling-related knowledge and perceived cycling confidence but was not sufficient to impact on behaviour in terms of frequency or an increase rate of cycling to school. Conversely, Johnson et al. [32] found the opposite in their study developed in the UK, where ‘Bikeability’ training was associated with an increase of frequency of cycling. Even the study of Bungum et al. [33] with a one-day intervention, showed that ACS intervention may provide an opportunity to enhance the proportion of youth who commuted actively, although it was acknowledged that the intervention was necessary more than once a day to create a healthy habit.

This clear disparity in the efficacy of cycling training and its impact on active commuting behaviour, coupled with the low rates of cycling to school in Spain, underline that it is important to act in this context. Consequently, the aims of this study were to: (1) assess the feasibility of a school-based cycling intervention in adolescents, and (2) analyse the effectiveness of a school-based cycling intervention on the rates of cycling to school, active commuting to school and barriers to ACS from adolescents.

## 2. Methods

### 2.1. Design Study and Sample

In this school-based randomized control trial, a random sample of six public secondary schools from three cities (Granada, Jaen and Valencia) in Spain was selected to participate in this study. In each city, there was an intervention group (hereinafter called cycling group) and a control group. The data were collected between 2019 and 2020 as part of the PACO (Pedalea y Anda al Cole/Cycling and Walk to School) Study. The PACO Study examines ACS in Spanish children and adolescents and several aims to develop interventions to promote adolescents’ ACS. The complete information of the recruitment, randomization process and procedure has been published elsewhere [34]. The PACO Study was approved by the Review Committee for Research Involving Human Subjects at the University of Granada (Reference: 162/CEIH/2016).

The sample initially recruited 150 adolescents (Figure 1) who were included in the intention-to-treat analysis. After application of the inclusion criteria, the final sample included in this study was 122 adolescents in the per-protocol analyses (cycling group, *n* = 60; and control group, *n* = 62) across the three cities. The cycling group participated in a school-based intervention to promote cycling to school within Physical Education (PE) sessions. In the current study, the inclusion criteria were: (a) adolescents from the third grade of secondary education (ages 14–15) (b) adolescents having completed both baseline and follow-up intervention questionnaires, and (c) those having attended at least 70% (three sessions of four) of the entire intervention (i.e., cycling group).

### 2.2. School-Based Intervention

The protocol for the school-based intervention used in this study has been published elsewhere [34,35], and is available online (http://profith.ugr.es/pages/investigacion/recursos/manualbici/, accessed on 3 August 2022). In addition, a brief explanation is presented in the section “Description of the School-Based Intervention”.

#### 2.2.1. Description of the School-Based Intervention

The school-based intervention is based in the Bikeability methodology [36]. The intervention was conducted in four PE sessions during one month (1 class per week).
➢1°. Theoretical session (60 min). The session included awareness about the benefits and usefulness of cycling as a mode of commuting in the city, and learning basic road safety rules, cycling safety equipment for both the rider and the bike, and cycling hand signaling in an urban context.➢2°. Closed circuit session (120 min. This session took place in the playground of the school in a traffic-free space. The session included correct helmet fitting, bicycle safety check before starting to ride, and fundamental cycling skills of starting off and pedaling, breaking safely, changing gears and hand signaling to change directions.➢3°. Urban circuit session (120 min). The participants used the knowledge and the skills learned in previous sessions in a real traffic context. The session included starting from the side of road (kerb), stopping on the side of road (kerb), overtaking a parked or slower-moving vehicle, lane changing, turning right and left and crossing a roundabout.➢4°. “Bicycle’s party” (120 min). The students had the opportunity to demonstrate what they had learned in previous sessions by becoming the teachers of a first grade secondary education group. The session included a circuit with several exercises based on knowledge and fundamental cycling skills learned in the previous sessions about urban cycling.

#### 2.2.2. Pilot Phase of the School-Based Intervention

First, a pilot phase was undertaken in the city of Granada, within PE sessions in a small sample of 14 students (not included in the RCT sample) from the third grade of secondary education in a private secondary school. This pilot phase studied the implementation of the intervention using different measures regarding the feasibility and the barriers perceived by the students, including: (1) observations of the research team during the session; (2) an interview with the PE teacher after the pilot intervention; (3) a focus-group with the students performed after the pilot intervention, and (4) a self-reported student questionnaire with information regarding the measures of enjoyment, usefulness, and potential improvements after each session.

The students stated that the third session (urban circuit session) was most liked, although they were afraid of the last activity (e.g., how to cross a roundabout safely). The PE teacher reported an intention to incorporate the intervention into the PE high school programme. In addition, he suggested adding more sessions to teach students who might be less experienced in the use of a bike and cycling on road with a bike. Both students and the PE teacher recommended that the fourth session (bicycle party) could be better organized regarding the planning of activities and their timing.

### 2.3. Measures

Several measures were used to address the objectives which were implemented at baseline, during the intervention, and at follow-up. The complete information has been described in detail elsewhere [35].

The measures used to address the feasibility of the school-based intervention were cycling knowledge, cycling skills (in a traffic-free area and on-road), and enjoyment, usefulness and improvements.
-Cycling knowledge. A self-reported questionnaire was completed by participants in the classroom at baseline and follow-up. The questions were about route safety rules, cycling hand signaling, and traffic. It consisted of a 20-item questionnaire with multiple-choice answers with three options and only one correct answer [35]. The score for each participant represented the number of correct answers.-Cycling skills in a traffic-free area. A cycling ad-hoc observational checklist in a traffic-free situation was completed by participants once during the intervention. The test was about cycling skills, including bike and hand signaling safely. It was composed of an 18-item checklist with dichotomy answers (yes/no), ranging from 0 points (lowest score indicating “It does not have the capacity to carry out the urban circuit”) to 18 points (highest score indicating “Unbeatable capabilities for the street circuit”). Further details about the observational checklists can be found in a previous publication [35].-Cycling skills on-road. A cycling ad-hoc observational checklist on road traffic situations was completed by participants once during the intervention. The tests were about cycling skills and signaling safely. It was a 22-item checklist with dichotomy answers (yes/no), ranging from 0 points (lowest score indicating “Low Cycling Capabilities”) to 22 points (highest score indicating “Expert Cyclist”).-Enjoyment, usefulness and improvements. A short questionnaire was completed by participants at the end of each of the four sessions during the intervention [35]. There were two questions about enjoyment and usefulness with a Likert scale of 5 points (5, “Totally agree”; 1, “Totally disagree”), and 1 question with an open answer regarding potential improvements.

The measures used to analyse the effect of the intervention were “active commuting to/from school” and “perceived barriers to active commuting to school”. These were collected at baseline and follow up intervention:
-Active commuting to/from school. A self-reported questionnaire (“Mode and Frequency of Commuting to and from School” questionnaire [37,38]) was completed by participants in the classroom at baseline and follow-up. The questions were about the latest weekly patterns of commuting to and from school. The possible answers were walking, cycling, car, motorbike, school bus, public bus, metro/train or other; only one option could be chosen. The participants were categorized as “active” if they reported walking or cycling as their usual mode of commuting and as “passive” if they answered car, motorbike, school bus, public bus, metro/train.-Perceived barriers to active commuting to school. A self-reported questionnaire “BATACE’s” (“Barreras en el Transporte Activo al Centro Educativo”, Spanish acronym) questionnaire [39,40] was completed by participants in the classroom at baseline and follow-up. The questions were about the barriers perceived concerning active commuting to school. The possible answers assessed using a Likert scale of 4 points to answer (4, “Totally agree”—higher perception of the barrier—; 1, “Totally disagree”—lower perception of the barrier—). A global index of the perceived barriers was calculated by the mean of the perceived barriers.

### 2.4. Statistical Analysis

The descriptive data of the participants are presented as frequencies (and percentages) for categorical variables and mean and standard deviation for continuous variables. Normality was assessed using the Kolmogorov-Smirnov test. Since the results showed that the age and cycling knowledge did not follow the normal distribution, these two variables were analyzed using non-parametric tests.

Differences between groups were calculated using Student’s *T* test and the U-Mann Whitney test for continuous variables (parametric and non-parametric test), and the chi-square test for categorical variables.

To analyze the changes in the dependent variables at baseline and follow-up of the intervention, differences were observed a comparison test of related samples, such as the Student *T*-test and non-parametric tests in those variables with free distribution (Wilcoxon, Signs and McNemar), both separately in the control group and in the cycling group. To establish the association between the dependent variables and the intervention, binary logistic regression models were performed. Differences at baseline and follow-up intervention were established as dependent variables and the intervention group variable was established as the independent variable for the analysis.

All analyses were undertaken using the statistical package SPSS for Windows version 23 (SPSS Inc., Chicago, IL, USA), and with a level of statistical significance of *p* < 0.05.

## 3. Results

The descriptive data of participants are presented in Table 1. The sample included 122 students, 49.5% boys, 50.5% girls) and their mean age was 14.26 ± 0.44 years old. A total of 66.3% of the students owned a bike and 96.8% of them did not cycle to/from school. The children’s mode of commuting to/from school was mostly active, and a percentage of 81.6% and 52.3% were active in the cycling and control group respectively (*p* = 0.002).

Figure 2 shows session attendance for the cycling group (*n* = 60). Mean attendance (number of participants) at the sessions was *n* = 47.5 out of 60.

Table 2 shows the differences in the perceived barriers between the cycling and control group at baseline. The results showed that the cycling group perceived fewer barriers to active commuting to school than the control group, although there were no differences between groups (*p* > 0.05).

Figure 3 presents the results of cycling knowledge, showing improvements of 2.02 points at follow-up compared to baseline in the cycling group (*n* = 44) (*p* < 0.001), 20 being the maximum score.

The Figure 4 presents the descriptive data of the cycling skills including scores for cycling skills in a traffic-free area (12.52 ± 3.54) and on-road cycling skills (10.94 ± 6.89), in the cycling group (*n* = 44).

Figure 5 shows data concerning enjoyment and usefulness of every session for the cycling group (*n* = 60). The mean of enjoyment was 4.60, and the mean of usefulness of the sessions was 4.78, both from a scale of 5 points.

Table 3 presents the changes in the mode of commuting to/from school and children’s perceived barriers to active commuting to school at baseline and follow-up for both groups (i.e., control and cycling). The ranking provides data on the comparison of participants at baseline and follow-up. The results, stratified by group, showed the negative ranks when the values at follow-up were higher than at baseline. The positive ranks indicated that the values at follow-up were lower than at baseline and a tie showed that there were no changes at follow-up, i.e., the same mode of commuting was maintained in the participants after the intervention. There were no significant differences (*p* > 0.05) in the changes of the mode of commuting between the baseline and follow-up. Regarding the barriers, there was only a change between the baseline and follow-up in the built environment (walk) barrier (*p* = 0.002) in the cycling group, with a positive rank of 4, negative rank of 17 and tie of 19. There were no significant differences in the change of barriers within the control group.

Table 4 shows three logistic regression models to observe the relationship between the intervention and the dependent variables. Changes in the mode of commuting (active/passive) at baseline and follow-up were not associated with the school-based intervention (Odds Ratio = 0.6, Confidence Interval = 0.13–2.70). Regarding perceived barriers to ACS, changes in the perceived barriers at baseline and follow-up (global index) (Odds Ratio = 0.56, CI = 0.21–1.47) and changes in the item “*Built environment (walk)*” at baseline and follow-up were not associated with the school based-intervention (Odds Ratio = 0.4, CI = 0.12–1.49) *p* > 0.05).

## 4. Discussion

The findings of this study showed that: (1) the school-based intervention might be feasible in the school context since cycling knowledge improved after the school-based intervention, the scores of cycling skills were medium-low and the attendance, enjoyment and usefulness of the sessions were high; (2) rates of cycling to school and active commuting to/from school did not change after the school-based intervention, and only the “Built environment (walk)” barrier in the cycling group was more perceived on follow-up. No association was found between the participation in the school-based intervention with the rates of cycling or active commuting to school and the perception of barriers to active commuting to school.

The present findings indicate that the proposed school-based intervention is feasible and implementable in the school context, and appropriate for use by teachers and researchers. Actually, cycling knowledge improved in the cycling group after the school-based intervention. Previous studies [41,42,43] showed similar results, where children improved their cycling knowledge scores after a cycle education program. In fact, another intervention study in the USA [44] found that a cycling program could improve up to 4 points from a maximum of 13 points on average in terms of cycling knowledge. It is necessary to emphasize that the fact of improving cycling knowledge does not necessarily result in a change in cycling behaviour [43], although it could be an incentive or address a barrier for people to cycle to school, as their confidence could be increased. In addition, the participants in the current study concluded that they liked the sessions and found them useful. The students’ enjoyment increases their learning potential, so it is crucial to develop interventions that are highly satisfactory and enjoyable [43]. It is necessary to highlight the importance of designing a useful cycling program for the students because it represents an opportunity to increase cycling to school [45]. We must mention that in the current school-based intervention there was only one data measurement for cycling skills during the school-based intervention, including both cycling skills assessments in a traffic-free area and in road traffic, but they were not comparable to each other. Consequently, we cannot determine if there was an improvement in cycling skills at follow up of the intervention, as other studies reported, indicating cycling skills improvements after a school-based interventions [21,43,46]. In addition, we must highlight that few participants in the current study did not use their own bicycle for the school-based intervention, because they did not have a bicycle, or it was not in a suitable condition for use. The research team and the school provided bicycles to the participants. If they could not access a bicycle of a suitable standard to cycle to school, this might affect the association between enhanced cycling skills and a change in the behaviour [47]. Mitchie et al. [48] suggested that to adopt and maintain a behaviour, the individual must have capability (C), opportunity (O) and motivation (M) for behaviour (B) (COM-B model of behaviour) to change. This intervention increased capability and motivation, but if participants did not have a bike they had no opportunity to cycle and would not change their behaviour.

There was no change in the rates of cycling and active mode of commuting to/from school at baseline and follow-up in both cycling and control groups. A previous school-based intervention of four sessions per week did not find change the children’s cycling to school after an intervention about cycling skills [29]. Similar results were found in the study of Groesz et al. [41] from Texas, which included 15 PE sessions, where increased cycling to school was not found but there was an increase in recreational cycling [41]. However, the study of Groesz et al. [41], found that cycling to school did not increase even though there was improved confidence in cycling [41]. In addition, the studies of Hatfield et al. [42] (eight sessions), Jones et al. [21] (five sessions) and Montenegro et al., [46] (eight sessions), showed increases in cycling to school after a school-based intervention. A potential explanation for these results may be that it is important to involve families in the interventions. It has been shown that family involvement in this type of interventions can be effective in promoting children’s PA [49]. The children’s cycling to school can be determined by parental attitudes and social norms, and household travel schedules [50,51]. However, previous study showed no increase tin he rates of cycling to school even when parents were involved [29], indicating that there are a broad range of factors that may underlie changes in the mode of commuting, such as the lack of cycle parking at schools [52,53] or the built environmental attributes of the school neighborhood [54]. It is well known that changing behaviours in our lifestyles is complex [55], and may be even more complicated in potentially new and dangerous situations such as cycling in an urban context. In addition, starting to cycle to school may require specific circumstances, such as both adolescents’ and parents’ consent, a suitable bike to cycle, living a bikeable distance between school and home and a safe route, among others.

In relation to changing the perceived barriers to active commuting to school, the results showed that there was only a significant change in the cycling group in terms of the built environment for walking. Consequently, the cycling group increased the perception of built environment (walk) as a barrier to actively commute. Regarding the other perceived barriers, there were no differences. A potential reason might be that the school-based intervention did not focus on the change of the perception of barriers directly. It was more focused on the cycling knowledge, cycling skills and cycling behaviour, and, consequently, improving the barriers maybe require another approach [29]. Despite this, it was expected that the perceived barriers would be reduced as the participants tested real situations in the school-based intervention (i.e., stay and manage traffic situations perceiving risks). However, the perception of the built environment as a barrier to walking to school increased. This may be caused by increase in awareness of the importance of a good built environment to safety commute to school [56]. It seems necessary to design interventions that focus attention on reducing barriers to active commuting. In the literature, we find that adolescents perceive different barriers from their parents [57], and yet parents are the main decision makers concerning commuting of their children [58]. In the case of parents, they reported barriers to active commuting as the distance between home and school, the built environment, traffic safety, crime-related safety, social support and physical and motivation barriers [59]. Similarly, children also reported convenience in addition to those identified by parents [60].

The topic of designing and implementing interventions to promote active commuting is still relatively new, especially in Spain, and it is worth keep elaborating and conducting cycling interventions to support a change in commuting behaviour, taking in account that a wide range of factors from an ecological perspective may have an influence (e.g., environmental, psychosocial or personal factors). It is also important to acknowledge the influence of context and culture [61] and social and cultural norms in a community. Clearly, it would be easier to develop school-based interventions in countries where there is already a “bicycle culture” and attitudes toward using bicycles is higher than in Spain [62,63]. For instance, other countries have specific national programs where children and young people learn to cycle safely, for example Belgium [29,64], the UK [65] and in Ireland [66]. In these countries cycling training in elementary/primary schools is a common approach. Consequently, interventions in countries where the cycling to school rates are as low, such as Spain, should focus at first instance on increasing knowledge, skills, and students and parents’ barriers towards active commuting to schools, and considering the role of the school in promoting active commuting as part of a wider agenda on a whole-school approach to physical activity promotion. Additionally, with the rise of physical activity interventions using co-production [67], there is an opportunity to co-create an intervention with and for young people to promote active commuting to school as a potential way forward to assist with these challenges. The interventions must be attractive to young people, since enjoyable interventions can be useful for participants to learn skills [43], and be focused on increasing the rate of cycling to school [45]. The adoption of co-production practices to develop and design interventions will contribute to addressing these factors.

This study has several limitations. First, we must mention some limitations within the intervention programme, such as the short duration, the lack of family involvement, the lack of a second assessment of cycling skills to compare these results, and the lack of the cycling knowledge in the control group. Moreover, the proposal was not included in the specific didactic unit of the curricula of the physical education teachers. Regarding the strengths of the study, the delivery of the intervention in school PE sessions (rather than an extra) and the design of the school-based intervention based on a randomized controlled trial performing in three cities with different contextual characteristics, must be highlighted.

## 5. Conclusions

We conclude that a school-based intervention might be a feasible approach to address active commuting in the school context, but some changes and modifications should be made. These are detailed in recommendations for future research and practice. Recommendations for the future are to continue with these promising interventions but include the participation of the young people, their families and other key stakeholders such as school management, local travel officers and policy makers to co-create multi-component interventions that address not only behaviour and knowledge, but challenge social norms, attitudes and physical infrastructure required to change behaviour in active commuting. In addition, a future effective initiative may include the promotion of cycling to school as a compulsory content in the curricula of Physical Education within national education policies.

## Figures and Tables

**Figure 1 ijerph-19-09626-f001:**
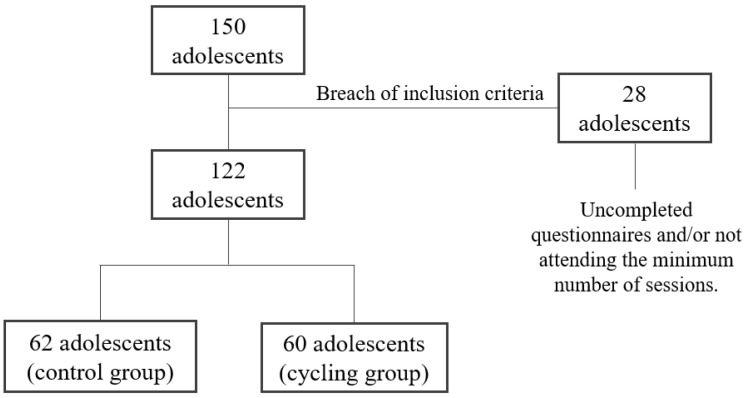
Flow chart of participants.

**Figure 2 ijerph-19-09626-f002:**
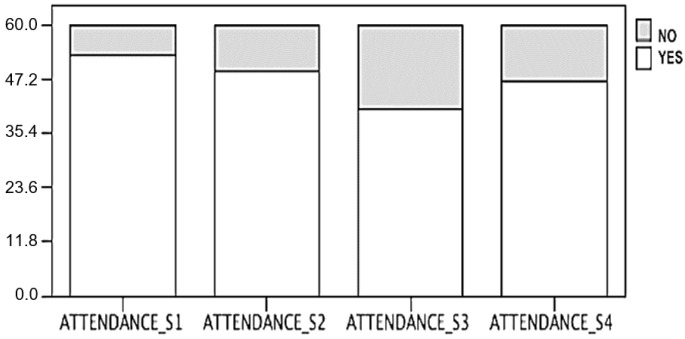
Attendance of the cycling intervention.

**Figure 3 ijerph-19-09626-f003:**
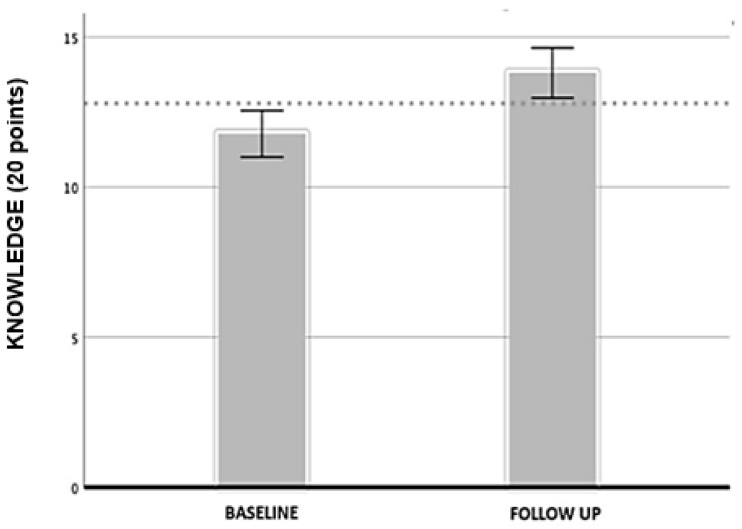
Changes on the cycling knowledge at baseline and at follow up on the cycling group.

**Figure 4 ijerph-19-09626-f004:**
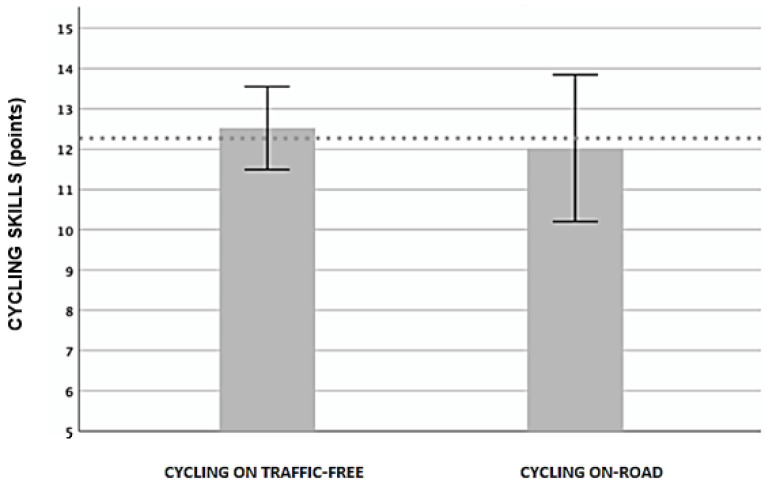
Descriptive data of the cycling skills in the cycling group.

**Figure 5 ijerph-19-09626-f005:**
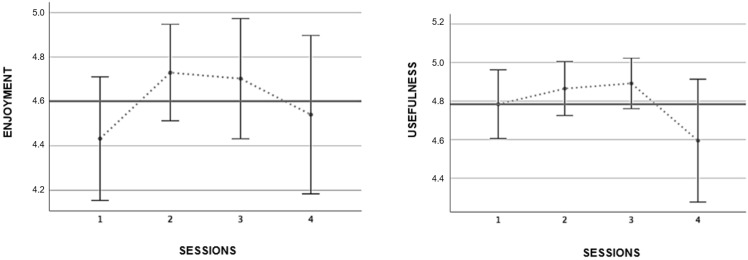
Enjoyment and usefulness of the cycling intervention. The bold line indicates the mean.

**Table 1 ijerph-19-09626-t001:** Descriptive data of the participants from the cycling and control group at baseline (intention to treat data).

	*All* *(n = 122)*	*Cycling Group* *(n = 60)*	*Control Group* *(n = 62)*	*p*
Children’s age *M* ± *(SD)*	14.26 ± 0.44	14.20 ± 0.40	14.32 ± 0.47	0.209
Children’s gender *n* (%)				
Boy	48 (49.5)	28 (54.9)	20 (43.5)	0.261
Girl	49 (50.5)	23 (45.1)	26 (56.5)	
Own bike *n* (%)	63 (66.3)	28 (57.1)	35 (76.1)	**0.008**
Commuting to/from school of children *n* (%)				
Active	63 (67.7)	40 (81.6)	23 (52.3)	**0** **.002**
Passive	30 (32.3)	9 (18.4)	21 (47.7)	
Cycling to/from school *n* (%)				
Cycling	3 (3.2)	3 (6.1)	-	0.950
Do not cycling	90 (96.8)	46 (93.9)	44 (100)	

Data in bold = Significant changes; *p*-value < 0.05; M ± (SD): Mean ± standard deviation; *n* (%): sample (percentage).

**Table 2 ijerph-19-09626-t002:** Descriptive data of perceived barriers by intervention group at baseline.

Perceived Barriers to ACS *n* (%)	*Cycling Group*				*Control Group*				*p*
	*Totally Disagree*	*Disagree*	*Agree*	*Totally Agree*	*Totally Disagree*	*Disagree*	*Agree*	*Totally Agree*	
Distance	27 (58.7)	8 (17.4)	3 (6.5)	8 (17.4)	19 (41.3)	7 (15.2)	8 (17.4)	12 (26.1)	0.210
Safety Traffic	20 (40.8)	12 (24.5)	9 (18.4)	8 (16.3)	18 (39.1)	13 (28.3)	9 (19.6)	6 (13)	0.953
Convenience	19 (38.8)	13 (26.5)	11 (22.4)	6 (12.2)	10 (21.7)	14 (30.4)	14 (30.4)	8 (17.4)	0.336
Built Environment	16 (32.7)	15 (30.6)	12 (24.5)	6 (12.2)	22 (48.9)	8 (17.8)	8 (17.8)	7 (15.6)	0.285
Crime Related Safety	21 (42.9)	18 (36.7)	8 (16.3)	2 (4.1)	17 (37)	24 (52.2)	5 (10.9)	-	0.275
Weather	37 (75.5)	5 (10.2)	4 (8.2)	3 (6.1)	30 (65.2)	7 (15.2)	7 (15.2)	2 (4.3)	0.574
Physical and Motivational Barriers	6 (12.2)	13 (26.5)	12 (24.5)	18 (36.7)	9 (19.6)	11 (23.9)	10 (21.7)	16 (34.8)	0.808
Built Environment (Walk)	27 (55.1)	12 (24.5)	8 (16.3)	2 (4.1)	22 (47.8)	14 (30.4)	9 (19.6)	1 (2.2)	0.810
Social Support (Walk)	26 (54.2)	4 (8.3)	7 (14.6)	11 (22.9)	18 (40)	8 (17.8)	7 (15.6)	12 (26.7)	0.434
Physical and Motivational Barriers (Walk)	22 (44.9)	17 (34.7)	5 (10.2)	5 (10.2)	23 (50)	10 (21.7)	6 (13)	7 (15.2)	0.538
Built Environment (Bike)	11 (22.4)	25 (51)	11 (22.4)	2 (4.1)	16 (34.8)	18 (39.1)	12 (26.1)	-	0.260
Social Support (Bike)	26 (54.2)	4 (8.3)	7 (14.6)	11 (22.9)	18 (40)	8 (17.8)	7 (15.6)	12 (26.7)	0.434
Physical and Motivational Barriers (Bike)	22 (44.9)	17 (34.7)	5 (10.2)	5 (10.2)	23 (50)	10 (21.7)	6 (13)	7 (15.2)	0.538
Global index	1 (2.2)	25 (55.6)	17 (37.8)	2 (4.4)	3 (6.8)	18 (40.9)	21 (47.7)	2 (4.5)	0.466

*n* (%): sample (percentage).

**Table 3 ijerph-19-09626-t003:** Changes in the mode of commuting to/from school and children’s perceived barriers to ACS at baseline and at follow up of the school-based intervention for both groups (per protocol data).

	* **Cycling Group** *				* **Control Group** *			
**MODE OF COMMUTING**								
	**Positive Ranks**	**Negative Ranks**	**Ties**	* **p** *	**Positive Ranks**	**Negative Ranks**	**Ties**	* **p** *
Active commuting to/from school	13	19	28	0.377	10	19	33	0.137
Cycling to/from school	3	1	56	0.625	2	2	58	1.000
	** *Cycling Group* **				** *Control Group* **			
**PERCEIVED BARRIERS TO ACS**								
	**Positive Ranks**	**Negative Ranks**	**Ties**	** *p* **	**Positive Ranks**	**Negative Ranks**	**Ties**	** *p* **
Distance	8	10	19	0.821	13	12	17	0.501
Safety Traffic	16	12	12	0.907	13	14	15	0.861
Convenience	11	11	18	0.573	15	13	15	0.674
Built Environment	11	11	18	0.813	10	12	19	0.892
Crime Related Safety	9	16	15	0.252	8	12	23	0.437
Weather	6	19	14	0.086	9	12	21	0.387
Physical and Motivational Barriers	17	9	13	0.100	12	10	20	0.829
Built Environment (Walk)	4	17	19	**0.002**	9	15	19	0.178
Social Support (Walk)	11	10	17	0.685	17	14	10	0.530
Physical and Motivational Barriers (Walk)	11	15	14	0.302	12	14	17	0.784
Built Environment (Bike)	8	17	15	0.066	9	14	20	0.263
Social Support (Bike)	11	10	17	0.685	17	14	10	0.530
Physical and Motivational Barriers (Bike)	11	15	14	0.302	12	14	17	0.784
Global index	6	14	20	0.061	4	10	29	0.225

Data in bold = Significant changes; *p*-value < 0.05.

**Table 4 ijerph-19-09626-t004:** Logistic regression of the differences in active commuting and perceived barriers at baseline and follow-up.

	Model 1: dv. Differences of Baseline—Follow-Up of Active Commuting to/from School	Model 2: dv. Differences of Baseline—Follow-Up of the Global Index	Model 3: dv. Differences of Baseline—Follow-Up of *“Built Environment (Walk)”*
	Odds Ratio (CI 95%)	Odds Ratio (CI 95%)	Odds Ratio (CI 95%)
*Cycling group* *	0.6 (0.13–2.71)	0.56 (0.21–1.47)	0.42 (0.12–1.49)

* Reference’s category: Control group; All *p* values were not significant; CI = Confidence interval; DV = Dependent Variable.

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
