# Peer review of "The Effect of a School-Based Intervention on Children’s Cycling Knowledge, Mode of Commuting and Perceived Barriers: A Randomized Controlled Trial"

_ijerph, 2022, doi:10.3390/ijerph19159626_

Round 1

Reviewer 1 Report

This study evaluated the effects of a school-based intervention program on active commute to school.  It is a topic worth investigating. However, there are significant issues of research design that preclude me from recommending this paper. 

Major issues: Research design

This paper is claimed to be a randomized controlled trial study. Even though the authors mentioned in the discussion section that there are multiple factors related to children’s travel behaviors, this study only managed to control participants’ age. Other factors such as built environment, school policies, parents’ attitudes towards biking to schools, and so on that are related to children’s travel behaviors were absent from the characteristics of matching controlling groups. It is unclear why those confounding factors were not controlled. 

Minor issues

This paper needs English editing to clear result reporting. 

Author Response

Comment

This study evaluated the effects of a school-based intervention program on active commute to school.  It is a topic worth investigating. However, there are significant issues of research design that preclude me from recommending this paper. 

 Major issues: Research design

 This paper is claimed to be a randomized controlled trial study. Even though the authors mentioned in the discussion section that there are multiple factors related to children’s travel behaviors, this study only managed to control participants’ age. Other factors such as built environment, school policies, parents’ attitudes towards biking to schools, and so on that are related to children’s travel behaviors were absent from the characteristics of matching controlling groups. It is unclear why those confounding factors were not controlled. 

Answer.

Thank you for your comment. We agree that several factors are related to children’s active commuting to school. Indeed, most of them are presented in the methodological manuscript about the PACO Study (please, find below the reference). However, in this study trial, the age, as well as the city and the type of school (public or private) were used to control the analyses.

“Chillón P, Gálvez-Fernández P, Huertas-Delgado FJ, Herrador-Colmenero M, Barranco-Ruiz Y, Villa-González E, Aranda-Balboa MJ, Saucedo-Araujo RG, Campos-Garzón P, Molina-Soberanes D, Segura-Díaz JM, Rodríguez-Rodríguez F, Lara-Sáncez AJ, Queralt A, Molina-García J, Bengoechea EG, Mandic S. A school-based randomized controlled trial to promote cycling to school in adolescents: The PACO study. International Journal of Enviromental Research and Public Health 2021, 18, 2066. doi: 10.3390/ijerph18042066.”

The reference is included at the methods of this manuscript

Minor issues

Comment.

This paper needs English editing to clear result reporting. 

Answer.

Thank you for your suggestion, the English writing has been reviewed.

Reviewer 2 Report

Dear authors, 
thanks for your work.
I really appreciate how you focused how people at young age perceive the barriers in order to transfer good attitudes in adulthood.
I also appreciate the "pilot phase" were you tested the whole study in a smaller group in order to perform the best strudy as you could.
Finally, the suggestion of improving the length of the intervention period is a good point to improve for further studies about the commuting to/from school among adolescents.

Author Response

Comments

Dear authors, 
thanks for your work.
I really appreciate how you focused how people at young age perceive the barriers in order to transfer good attitudes in adulthood.
I also appreciate the "pilot phase" were you tested the whole study in a smaller group in order to perform the best study as you could.
Finally, the suggestion of improving the length of the intervention period is a good point to improve for further studies about the commuting to/from school among adolescents.

Answer

Thank you for your comments.

We have tried to design this intervention looking for a way to modify their perception, so that, in the long term, it becomes established.

Regarding the duration of the intervention, although the literature shows several options, we agree that it would be important to increase the number of sessions in order to obtain greater success.

Reviewer 3 Report

Thank you very much for the opportunity to read and comment on the article: "The effect of a school-based intervention on children's cycling knowledge, mode of commuting and perceived barriers: a randomized controlled trial", it is a well-structured work, however, it has some methodological errors.

General:

The work is carried out by 12 authors, I find it strange how many authors were responsible for reviewing the article and did not detect gross errors in writing a scientific article. can you detail the function of each one?

I believe that a program of this nature should not be implemented in physical education classes, but a specific curriculum enrichment program should be created.

Why wasn't it done?

Do you not consider it to be a limitation of the study?

abstract:

I think that the abstract should have a first introductory paragraph.

It must have at least one more keyword.

Methods:

Lines 94-97: create a flow diagram of participants

2.2.1 (3º) - How was the "Urban Circuit" created and controlled in the three different cities?

Was it different from city to city?

2.2.2 PRIVATE SECONDARY SCHOOL. 

Why was this study carried out in private and not in public schools? Would it be different? will be a limitation?

REF BIBLIO 35:Is the questionnaire validated? For this population?

Questionnaires were made in what language? If they were in Spanish, are they all translated and have they been culturally adapted?

If the questions were in English. What is the students' level of English?

There are questionnaires that do not have the name and/or Bibliographic reference (p.e. page 170 "SHORT QUESTIONNAIRE").

Road cycling skills.

These types of skills may differ from individual to individual depending on the obstacles of cities, and the expert in one city may not be the expert in another...

line 216: 47.7% or 52,3% 

Table 2: Why are some variable titles in bolt and another not?

Figures 1 and 2: Replace MEAN on axis, with the variable name.

p=???

Figure 3: I consider it can be written in the text at the beginning of point 3.

line 243: The author mixes the description of figure 3 with figure 4

Figure 3: p=??? » Sessions to sessions..

Quite "mean"

Table 3: I need the ACSs to be well-defined.

Can you explain better the "TIES"

Table 4: Put the equations of each regression model.

Discussion:

Based on lines 306-311: Tell me what is this study for and why should it be published?

The study has 4 sessions, no discussion study has 4 sessions, with significant effects, neither with 8 or 12 sessions. Was there no prior literature review (12 authors)?

This type of study should always include the family. Why wasn't it done?

The authors speak of cultural identity influence. The discussion should have a paragraph on the application of methodologies of a political nature, which broke down barriers and which certainly fostered a change in cultural habits. As in countries mentioned in the discussion and others not mentioned.

Conclusions

The conclusions should be more summarized responding only to the proposed objectives.

Author Response

Comments

Thank you very much for the opportunity to read and comment on the article: "The effect of a school-based intervention on children's cycling knowledge, mode of commuting and perceived barriers: a randomized controlled trial", it is a well-structured work, however, it has some methodological errors.

General:

The work is carried out by 12 authors, I find it strange how many authors were responsible for reviewing the article and did not detect gross errors in writing a scientific article. can you detail the function of each one?

Answer

Thank you for your comment.  The author contributions has been: Conceptualization, M.J. A.-B., F.J. H.-D. and P.C.; funding acquisition, P.C.; methodology, M.J. A.-B., F.J. H.-D., M. H.-C., J.M.-G. A. Q., D. C., and P.C.; formal analysis, M.J. A.-B.; data curation, M.J.A.-B., P. G.-F., R.G.S.-A., D. M.-S., P. C.-G.; writing original draft preparation, M.J. A.-B., F.J. H.-D. and P.C.; writing—review and editing, M.J. A.-B., F.J. H.-D., P. G.-F., R.G.S.-A., D. M.-S., P. C.-G., M. H.-C., A.J. L.-S., J.M.-G. A. Q., D. C., and P.C. All authors have read and agreed to the published version of the manuscript.

Moreover, the authors did not only participate in the writing and review, but they have contributed to the development of the intervention at the three different cities where the program intervention was carried out. So this study could not have been done without the contribution of each author.

Comment:

I believe that a program of this nature should not be implemented in physical education classes, but a specific curriculum enrichment program should be created.

Why wasn't it done?

Do you not consider it to be a limitation of the study?

Answer:

The content of this intervention developed a manual (Salto C, Aranda-Balboa MJ, Gálvez-Fernández P, Herrador-Colmenero M, Chillón P. Proyecto de innovación educativa para la eso: “manual de intervención bikeability”. Habilidad motriz: Revista de ciencias de la actividad física y del deporte. 2019(52):12-38.) that has been created to promote the implementation of cycling and road safety education in the physical education curriculum. However, we agree with the researcher that it should be important to create a program specifically linked to the curriculum. Actually, part of our research group is now working with the ministry about the inclusion of this content in the Spanish physical education curriculum.

Comment:

abstract:

I think that the abstract should have a first introductory paragraph.

Answer:

Thank you for your comment. We have added a new sentence in the begin of the paragraph.

The low rates of active commuting to/from school in Spain, especially by bike and the wide range of cycling interventions in the literature show that it is necessary to research it.”

Comment:

It must have at least one more keyword.

Answer:

Thank you for your suggestion. The expression “school” was included as keyword.

Comment:

Methods:

Lines 94-97: create a flow diagram of participants

Answer:

Thank you for your comment. The figure was included at method.

Figure 1. Flow chart of participants.

Comment:

  • (3º) - How was the "Urban Circuit" created and controlled in the three different cities?

Answer:

Thank you for your comment. Each "Urban Circuit" was created by the same cycling expert and was implemented by the same expert. Indeed, this expert considered the better environment to prove each of the skills that need to be assessed and the different activities to be carried out to achieve them.

To develop the urban circuit, the first step was to use Google maps to design a circuit in the vicinity of the school. Secondly, the expert moved to the city to check it on the ground, in order to minimise risks and avoid any mishaps during the practice.

Comment:

Was it different from city to city?

Answer:

Thank you for your comment. All “Urban Circuits” included the same activities and distance. The only difference is the geographical position.

Comment:

2.2.2 PRIVATE SECONDARY SCHOOL. 

Why was this study carried out in private and not in public schools? Would it be different? will be a limitation?

Answer:

Thank you for your suggestion. The study was carried out at six public schools in Spain as it is described at methods.

Methods: “In this school-based randomized control trial, a random sample of six public secondary schools from three cities (Granada, Jaen and Valencia) in Spain was selected to participate in this study.”

Only the pilot phase was carried out at a private school due to convenience because of the relation stablished previously with the physical education teachers.

The results in both institutions (public and private) have been similar so we do not believe that there is a big difference in one or the other. In Spain, there is a figure of private school that receives public funding and present similar characteristic in terms of socioeconomic status and location that public schools.

Comment:

REF BIBLIO 35:Is the questionnaire validated? For this population?

Answer:

Thank you for your comment. If you are referring to the questionnaire developed for the assessment of road safety knowledge (Test de evaluación de normas de circulación y comportamiento ciclista), it is the only that has not been validated.

It is a series of basic questions to check the participants' previous knowledge and see if it improves after the intervention. It is like a physical education exam about this specific content.

Comment:

Questionnaires were made in what language? If they were in Spanish, are they all translated and have they been culturally adapted?

Answer:

Thank you. All the questionnaires used in this study was created and validated in Spanish young population.

Comment:

If the questions were in English. What is the students' level of English?

Answer:

Thank you. As we answered in the previous question all the questionnaires were developed in Spanish.

Comment:

There are questionnaires that do not have the name and/or Bibliographic reference (p.e. page 170 "SHORT QUESTIONNAIRE").

Answer:

Thank you for your suggestion. The bibliographic reference was included at text.

Comment:

Road cycling skills.

These types of skills may differ from individual to individual depending on the obstacles of cities, and the expert in one city may not be the expert in another...

Answer:

Thank you for your comment. We agree with the reviewer comment, as the skills depend on the individual level. To minimise this difference, the second session was held within the school where different skills needed to drive in the city are shown, to have a group of participants with a similar level. In addition, the urban circuits are designed by an expert in the field. Specifically for this study, it was the same expert who designed and tested all the circuits. On the other hand, the researchers who carry out the evaluation are trained through a course in order to be as objective as possible.

Comment:

line 216: 47.7% or 52,3% 

Answer:

Thank you for your comment. The text was corrected.

“The children’s mode of commuting to/from school was mostly active, and a percentage of 81.6% and 52.3% were active in the cycling and control …”.

Comment:

Table 2: Why are some variable titles in bolt and another not?

Answer:

Thank you for your comment. We have modified the titles.

Comment:

Figures 1 and 2: Replace MEAN on axis, with the variable name.

Answer:

Thank you for your comment. We have replaced by the variable name.

Comment:

p=???

Answer:

Please, could you specify about this comment?

Comment:

Figure 3: I consider it can be written in the text at the beginning of point 3.

Answer:

Thank you for your suggestion. We have changed the figure 3 after table 1.

Comment:

line 243: The author mixes the description of figure 3 with figure 4

Answer:

Thank you for your comment. We have separated the description of each figure.

Comment:

Figure 3: p=??? » Sessions to sessions..

Answer:

Thank you for your comment. The Figure 3 (now, figure 1) show the attendance of each session, only. There is no data about the significance values due to it is not necessary to achieve the objective of the work.

Comment:

Quite "mean"

Answer:

Thank you for your comment. The mean attendance presents of Figure 3 (now, 1) refers to the attendance of participants (47.5 of participants from 60 participants in total). It is a good number of the total of cycling group.

Comment:

Table 3: I need the ACSs to be well-defined.

Answer:

Thank you for your comment. ACS means active commuting to/from school, where it is included walk or cycle. And cycling to/from school included cycling mode, only.

The definition it is included at introduction (lines 40 – 41).

Active commuting to and from school (ACS) can be a routine behaviour that enables pupils to be more physically active during their day, walking or cycling.”

Comment:

Can you explain better the "TIES"

Answer:

Thank for your suggestion. We have included a sentence to explain this concept.

“… and a tie showed that there were no changes at follow-up, i.e. the same mode of commuting is maintained in the participants after the intervention…”.

Comment:

Table 4: Put the equations of each regression model.

Answer:

Thank you for your suggestion.  To analyse the data, we have used a statistic software, which does not give you an equation as such, what is shown are the dependent and independent variables.

Comment:

Discussion:

Based on lines 306-311: Tell me what is this study for and why should it be published?

Answer:

Thank you for your comment. Although a limitation of this study, as indicated in this section of the discussion, is the short period of the intervention, it is necessary to remark that this study is a good starting point for future research. Through this study, we have found that there is a need to increase the duration of such an intervention and to include more sessions in order to be able to compare skills. Therefore, it is necessary to publish it so that other researchers can take into account everything that has been done in this study.

Comment:

The study has 4 sessions, no discussion study has 4 sessions, with significant effects, neither with 8 or 12 sessions. Was there no prior literature review (12 authors)?

Answer:

Thank you for your comment. Even when the number of session were 4, it corresponded to 7 physical education school session over 4 weeks.The literature was reviewed showing that a study of 5 sessions (Jones et al., 2017), Hatfield et al., (2019) or Montenegro et al., (2015) with 8 sessions each, among others, presenting increased cycling at school after a school-based intervention. Even a study of 1 session found positive results after the intervention (Bungum et al., 2014).

However, we concluded after the intervention that it is important to increase the number of sessions.

Comment:

This type of study should always include the family. Why wasn't it done?

Answer:

Thank you for your comment. The family was included in the PACO Study.  The family completed a questionnaire and several data were collected that is being currently analysed. All information about the study design is published at:

“Chillón P, Gálvez-Fernández P, Huertas-Delgado FJ, Herrador-Colmenero M, Barranco-Ruiz Y, Villa-González E, Aranda-Balboa MJ, Saucedo-Araujo RG, Campos-Garzón P, Molina-Soberanes D, Segura-Díaz JM, Rodríguez-Rodríguez F, Lara-Sáncez AJ, Queralt A, Molina-García J, Bengoechea EG, Mandic S. A school-based randomized controlled trial to promote cycling to school in adolescents: The PACO study. International Journal of Enviromental Research and Public Health 2021, 18, 2066. doi: 10.3390/ijerph18042066.”

However, in this study, the family was not involved in the intervention, but it is a good idea to included them in the design. Even thought, there is a previous study that did not increase the rates of cycling to school even when parents were involved (Ducheyne et al., 2014).

Comment:

The authors speak of cultural identity influence. The discussion should have a paragraph on the application of methodologies of a political nature, which broke down barriers and which certainly fostered a change in cultural habits. As in countries mentioned in the discussion and others not mentioned.

Answer:

Thank you for your suggestion. We have included information about this topic at discussion section (line 398 – 416).

It is also important to acknowledge the influence of the context and culture [63] and social and cultural norms in a community. Clearly, it would be easier to develop school-based interventions in countries where there is already a “bicycle culture” and attitudes toward using bicycles is higher than in Spain [64, 65]. For instance, other countries have specific national programs where children and young people learn to cycle safely, for example Belgium [29, 66], the UK [67] and in Ireland [68]. In these countries cycling training in elementary/primary schools is a common approach. Consequently, interventions in countries where the cycling to school rates are as low, as Spain, should focus at first instance on increasing knowledge, skills and students and parents’ barriers towards active commuting to schools, and considering the role of the school in promoting active commuting as part of a wider agenda on a whole school approach to physical activity promotion. Additionally, with the rise of physical activity interventions using the co-production [69], there is an opportunity to co-create an intervention with and for young people to promote active commuting to school as a potential way forward to assist with these challenges. The interventions must be attractive to young people, since enjoyable interventions can be useful for participants to learn skills [43] and be focused on increasing the rate of cycling to school [45]. The adoption of co-production practices to develop and design interventions will contribute to address these factors.”

Comment:

Conclusions

The conclusions should be more summarized responding only to the proposed objectives.

Answer:

Thank you for your suggestion. The conclusion was summarized.

Reviewer 4 Report

Cycling as a form of physical activity is becoming more and more popular. Developing a bicycle-friendly infrastructure has become the goal of many governmental and non-governmental organizations. Popularization of the use of the bicycle as a means of communication, included in the school physical education curriculum, has its justification.

It would also be worth informing the reader how during the intervention in physical education lessons access to the bicycle was ensured for students who did not have it. I did not find an indication in the article that the lack of a bicycle was an exclusion from belonging to the experimental group.

In the abstract, the authors write that 122 students from Granada participated in the study. In the next sentence there is information about the research in Jaén and Valencia. Were studies conducted in each of these cities?

I have some doubts about how to assess the level of cycling skills.

It should be clearly stated here that this is the self-evaluation of the students. There is no information about the fact that the researchers verified the opinions of young cyclists. The authors themselves indicate this limitation at the end of the article

No data labels on the bars of graphs 1-3 - including such information will make the information more readable.

Author Response

Comments:

Cycling as a form of physical activity is becoming more and more popular. Developing a bicycle-friendly infrastructure has become the goal of many governmental and non-governmental organizations. Popularization of the use of the bicycle as a means of communication, included in the school physical education curriculum, has its justification.

It would also be worth informing the reader how during the intervention in physical education lessons access to the bicycle was ensured for students who did not have it. I did not find an indication in the article that the lack of a bicycle was an exclusion from belonging to the experimental group.

Answer:

Thank you for your suggestion. The complete information about the complete process of access to material and bicycle as included at PACO Study (Chillón et al., 2021). Also, there is some explanation at this paper (discussion: line 339 – 344).

“…In addition, we must highlight that few participants in the current study did not used their own bicycle for the school-based intervention, because they did not have, or it was not in a suitable condition for uses. The research team and the school contributed to provided bicycles to the participants. If they cannot access a bicycle of suitable standard to cycle to school, it might affect the association between enhanced cycling skills and a change in the behaviour [48]…”

Comment:

In the abstract, the authors write that 122 students from Granada participated in the study. In the next sentence there is information about the research in Jaén and Valencia. Were studies conducted in each of these cities?

Answer:

 Thank you for your comment. the information in the abstract includes that there are 3 participating cities, which have a total of 122 participants.

 “…A total of 122 adolescents from Granada, Jaén and Valencia (Spain) participated in the study…”

Comment:

I have some doubts about how to assess the level of cycling skills.

It should be clearly stated here that this is the self-evaluation of the students. There is no information about the fact that the researchers verified the opinions of young cyclists. The authors themselves indicate this limitation at the end of the article.

Answer:

 Thank you for your comment. The complete information about assessment as included at Salto et al., 2019, also there is a description of the assessment on the Methods of this paper.

 The assessment of the skills is carried out by an expert using worksheets (Salto et al., 2019). Students only evaluate the performance of the expert (monitor) and the sessions.

 Regarding the limitation mentioned in the paper, it refers to the fact that it is not possible to compare the expert's first assessment with a follow-up after the intervention.

 Comment:

No data labels on the bars of graphs 1-3 - including such information will make the information more readable.

Answer:

 Thank you for your suggestions. The labels have been included.

Round 2

Reviewer 3 Report

Dear authors, thank you very much for this excellent moment of sharing and learning, I confess that when I read the article for the first time, my intention was to REJECT it. However, the excellent review of the problems raised, the acceptance of them, and the respective corrections led me to now consider the article for publication. I consider the article much better now. As well as my knowledge about the topic presented.

Congratulations